**Data Availability Statement:** All relevant data are within the manuscript and its Supporting Information files.

**Funding:** This research was supported by the National Natural Science Foundation of China under

# Energy saving strategy of cloud data computing based on convolutional neural network and policy gradient algorithm

**Dexian Yang ⓘ \*, Jiong Yu, Xusheng Du, Zhenzhen He, Ping Li**

School of Information Science and Engineering, Xinjiang University, Urumqi, China

\* yangdexian@stu.xju.edu.cn

## Abstract

Cloud Data Computing (CDC) is conducive to precise energy-saving management of user data centers based on the real-time energy consumption monitoring of Information Technology equipment. This work aims to obtain the most suitable energy-saving strategies to achieve safe, intelligent, and visualized energy management. First, the theory of Convolutional Neural Network (CNN) is discussed. Besides, an intelligent energy-saving model based on CNN is designed to ameliorate the variable energy consumption, load, and power consumption of the CDC data center. Then, the core idea of the policy gradient (PG) algorithm is introduced. In addition, a CDC task scheduling model is designed based on the PG algorithm, aiming at the uncertainty and volatility of the CDC scheduling tasks. Finally, the performance of different neural network models in the training process is analyzed from the perspective of total energy consumption and load optimization of the CDC center. At the same time, simulation is performed on the CDC task scheduling model based on the PG algorithm to analyze the task scheduling demand. The results demonstrate that the energy consumption of the CNN algorithm in the CDC energy-saving model is better than that of the Elman algorithm and the ecoCloud algorithm. Besides, the CNN algorithm reduces the number of virtual machine migrations in the CDC energy-saving model by 9.30% compared with the Elman algorithm. The Deep Deterministic Policy Gradient (DDPG) algorithm performs the best in task scheduling of the cloud data center, and the average response time of the DDPG algorithm is 141. In contrast, the Deep Q Network algorithm performs poorly. This paper proves that Deep Reinforcement Learning (DRL) and neural networks can reduce the energy consumption of CDC and improve the completion time of CDC tasks, offering a research reference for CDC resource scheduling.

## 1. Introduction

With the fast progress of Information Technology (IT) and Cloud Technology, data centers are the basis for supporting business development and the only way for enterprise development [1]. As the social economy develops rapidly, the economy and market are also accelerating. The construction of data centers has received significant attention in market

Grant Nos. 61862060. The funders had no role in study design, data collection and analysis, decision to publish, or preparation of the manuscript. There was no additional external funding received for this study.

**Competing interests:** The authors have declared that no competing interests exist.

development. The data center's unreliability and low operating efficiency will inevitably affect data processing, which is not conducive to the development of enterprises [2]. The increase in large data centers is closely related to Cloud Data Computing (CDC). Currently, CDC centers are related to the Infrastructure-as-a-Service level and mainly focus on supply space [3]. Building small and medium data centers is difficult because of strict equipment requirements and high unit costs. Economic changes have led to a shortage of resources for small and medium-sized enterprises [4]. However, judging from the actual operation of the data center, the energy consumption cost of the data center remains high, especially the considerable power consumption. It gradually increases the burden on the increasingly high IT costs and fails to meet the development requirements of low energy consumption in today's society [5].

In the development of today's society, increasing attention is paid to energy utilization. In particular, the principle of energy saving and emission reduction should be adhered to in the design, construction, and development of data centers [6]. More than half of the energy that powers data centers is now was ted, including water, electricity, labor, and building space [7]. According to reliable statistics, the use of servers has increased six times, and storage capacity has increased 69 times in the last decade. 66.00% to 72.00% of enterprises plan to expand their data centers within 1 to 2 years [8]. The energy consumption of CDC centers mainly comes from IT equipment, such as server equipment, network interconnection equipment, and power supply equipment that constitute the bulk of cloud data centers. The energy consumption of temperature control equipment, such as air conditioners and fans required for the cooling system of the server room and the energy consumption of other auxiliary equipment [9]. The energy consumption of a data center should be considered at multiple levels. It is imminent to save energy and reduce emissions in data centers, considering various factors, domestic power supply capacity constraints, and the comprehensive factors of a green energy-saving environment [10]. The cloud computing platform gradually shows a proliferation of real-time characteristics in data. The mechanism of data deployment on the cloud computing platform mainly focuses on improving the efficiency of data access, the reliability of data storage space and the consistency of data control, and the regularity of accessing data, which leads to insufficient energy consumption [11]. Therefore, energy saving and emission reduction of data centers can be promoted by applying advanced technologies in the cloud context.

The most suitable CDC energy-saving strategy is designed to solve the problem of high energy consumption and consumption in CDC centers and realize secure, intelligent, and visualized energy consumption management. Promoting the innovation and development of the CDC industry is of great significance to the country's technological development, soft power improvement, and data security assurance. It is essential to strengthen the research on carbon peaking and carbon neutrality in the field of CDC and formulate cloud services and related carbon emission accounting methods as soon as possible. Meanwhile, it is necessary to encourage cloud service providers to actively carry out low-carbon technology and innovation research on cloud services and use CDC technology to empower enterprises and the world. This article paper discusses the theory of Convolutional Neural Networks (CNNs) and designs an intelligent energy-saving model based on CNN given the variable energy consumption, load, and power consumption of CDC data centers. Secondly, the core idea of the policy gradient (PG) algorithm is discussed. A CDC task scheduling model is designed based on the PG algorithm to ameliorate the uncertainty and volatility of the CDC scheduling tasks. Finally, the performance of different neural network models in the training process is analyzed with the total energy consumption and load balancing optimization in CDC as the research objects. Meanwhile, an experimental simulation is carried out on the CDC task scheduling model based on the PG algorithm for demand analysis.

Under the general trend of energy saving, establishing a perfect data center energy monitoring mechanism is a prerequisite for energy planning and energy management. Traditional hardware-based detection generally refers to the direct measurement of power consumption of the system under test through external power measurement devices or customized collectors. This approach is feasible in small-scale data centers, but the biggest drawback is that it does not meet the need for low-cost and easily scalable monitoring. In contrast, software-based energy consumption monitoring mechanisms can enable multi-granular, highly scalable monitoring systems cost-effectively. They are well suited for the complex, heterogeneous, and frequently scalable equipment environments in CDC centers. A CDC task scheduling model with a policy gradient algorithm is innovatively proposed to achieve energy-saving estimation and prediction to explore the application and generalization capability of the energy-saving model on increasing data sets. The algorithm reported here has good convergence in the scenario of high heterogeneity and task volatility of CDC clusters. It dramatically improves the command response time ratio of tasks, promotes cluster load balancing, and realizes the computing energy-saving strategy for cloud data centers in terms of time and device energy consumption.

## 2. Literature review

### 2.1 Research on energy-saving scheduling under cloud background

Yan et al. (2019) conducted simulation tests on traditional energy consumption measurement models and virtualized energy consumption models to solve the problem of high energy consumption in CDC data centers. They found that traditional energy models usually capture server power consumption through external power measurement devices or custom harvesters. This approach is suitable for small, modular, homogeneous data center environments. The virtualized energy consumption model has the advantages of low cost, easy expansion, and more granularity. At the same time, it can realize data collection and prediction at the virtual machine and container level [12]. Tahir et al. (2020) established a virtual integration mechanism in the CDC center, using virtual machines to enable the CDC data computing center to have efficient resource scheduling [13]. Hao et al. (2020) performed task prediction on data centers via K-means and Extreme Learning Machines [14]. Wu et al. (2019) analyzed the energy consumption of servers with different configurations in the data center through the time-series neural network model to find the best low-energy server configuration method [15]. Shen et al. (2022) studied the multi-granularity energy consumption model of the CDC data center, compared the energy consumption models of components and programs, and proposed a model with high accuracy, low cost, good generalization ability, high portability, and robust energy consumption [16]. Ebadi et al. (2019) divided the information and communication facilities of CDC data centers into different energy consumption units. The energy consumption statistics of different facility units suggested that the energy consumption of servers and storage devices was 40% higher than that of network devices, which is about 35% higher than the power consumption of the power supply unit. In addition, the energy consumption of the network equipment and the power supply unit is lower, so the energy consumption of the information and communication facilities in the CDC data center is mainly concentrated in the server and storage equipment [17]. Shah et al. (2022) used the Central Processing Unit (CPU) load index to estimate the power consumption of the entire data center server. The authors estimated the CPU load index for compute-intensive as well as hard disk and memory-intensive. The research results indicated that the energy consumption of the CPU was 58% during the whole load process [18]. Zeng et al. (2021) believed that CPU energy consumption is related to the CPU's working frequency in a study of the influencing factors of

CPU energy consumption. When each CPU thread is fully loaded, the higher the CPU's working frequency, the greater the energy consumption [19]. Huang et al. (2021) estimated the energy consumption of the data system CPU based on the CPU utilization in the computational research on energy consumption, where the energy consumption of the CPU is proportional to the CPU utilization [20]. Lin et al. (2020) used an exponential power function to study the energy consumption of popular servers from 2007 to 2010. They found that a functional model with an exponent of 0.75 could express these servers' energy consumption performance [21].

## 2.2 Research on task scheduling in the cloud background

There are also many task scheduling studies on the CDC platform. Chen et al. (2020) optimized task scheduling in the CDC environment to meet users' needs and reduce the task completion time [22]. Khorsand et al. (2020) analyzed the task scheduling optimization model in the CDC environment from the perspective of operators. Energy consumption can be reduced by improving the utilization of cluster resources in data center facilities on the premise of ensuring server load balance [23]. At the same time, task scheduling in the cloud environment can also be realized through algorithms. Ghobaei-Arani et al. (2020) used the Moth Flame Optimization algorithm to assign the optimal set of tasks to the data center nodes of CDC to minimize the total execution time of tasks [24]. Munir (2019) handled the movement of uncertain task clusters in the CDC environment online management method based on Mobile Edge Computing and Reinforcement Learning (RL), thereby achieving optimal task movement management [25]. Nassar (2019) combined RL and the greedy algorithm for crowd-aware task scheduling in mobile social networks. The optimal combination of the algorithms improved the perception efficiency and saved energy consumption [26]. Dong et al. (2020) used the RL-Task Scheduling algorithm to schedule tasks with a priority relationship between dynamic task scheduling and cloud servers. The research showed that the algorithm minimizes the task execution time [27]. Wang (2019) studied the scheduling strategy of highly heterogeneous tasks in the cloud environment. They also improved the action selection strategy to Boltzmann action selection strategy on the Deep-Q-Network (DQN) algorithm, which improved the inquiry ability of the $\varepsilon$-greedy algorithm [28]. Table 1 summarizes the characteristics of domestic and foreign scholars' studies.

To sum up, scholars have studied energy-saving scheduling or task scheduling in the CDC background from the energy consumption model and the energy consumption of the data center CPU. Traditional energy-efficient task scheduling algorithms such as round-robin scheduling and ant colony optimization have been extensively used in many cloud computing systems. Due to the dynamic nature of workloads, most existing online energy-saving algorithms are based on heuristics and rely heavily on historical experience. They can solve the resource scheduling problem effectively. However, once the scheduling target or resource pool changes, the designed heuristics cannot be used due to their static nature, and the designer

**Table 1. Characteristics of domestic and foreign scholars.**

| Research Technique | Research Strengths |
|---|---|
| Dynamic consolidation of virtual machines | Reducing total energy consumption |
| Deep Reinforcement Learning | Low latency |
| Data balanced partitioning | Load leveling |
| The research method reported here | Reduces computing power consumption, guaranteeing the quality of service, and improving resource utilization |

needs to redesign the scheduling algorithm for the new conditional environment. However, there is no research on the low energy consumption and energy-saving strategies of task scheduling of CDC data centers in the cloud environment. Machine Learning models can predict future load demand and scale resources as needed to provide excellent Quality of Service and user experience. These techniques are valuable for service providers to offer reliable services and maintain market leadership. In addition, neural networks and policy gradient algorithms, a machine learning branch, have evolved rapidly in recent years. This technology uses monitoring and feedback data to continuously optimize and improve policies and obtain optimal decisions through continuous intelligence interaction with the environment. This adaptive approach adjusts the agent's decisions and policies according to the current requirements, workload, and the underlying system's state. Therefore, the resource allocation policies for these tasks are to be intelligent for the modern and changing cloud data energy-saving task requirements and cloud environments. This work combines Deep Reinforcement Learning (DRL) and neural networks to reduce the energy consumption of task scheduling in CDC.

## 3. Research methodology

### 3.1 CNN theory

This work designs an intelligent energy-saving model based on CNN according to the characteristics of the CDC center with variable energy consumption, load, and power consumption [29]. Besides, a scheduling model of cloud data computing tasks based on the policy gradient algorithm is designed by the policy gradient algorithm for the uncertainty of CDC scheduling tasks and volatility [30].

CNN is a Feedforward Neural Network with neurons that can respond to a part of the surrounding units within the coverage area. It has been used in computer vision for a long time. Its core is "convolution and pooling" operations. A CNN is a multi-layer perceptron designed for recognizing two-dimensional shapes (such as images), consisting of an input layer, a hidden layer (convolutional layer and pooling layer), and an output layer. The hidden layer can have many layers; each layer consists of one or more two-dimensional planes; each plane consists of multiple independent neurons. A neuron in the convolutional layer of a CNN is only connected to some of its neighbors. Unlike ordinary neural networks, CNNs contain feature extractors consisting of convolutional and pooling layers [31]. Fig 1 presents the CNN structure.

Fig 1 shows a CNN model of a computer data center. The feature is input and then reaches the convolution layer. The convolution layer performs convolution through the convolution kernel and the 3×3 data to extract the local features of the computer data center. The pooling layer mainly reduces the computational complexity by reducing the dimension of the output matrix of the previous layer, extracting the practical features of the computer data center. Then, the fully connected layer realizes the connection of data features. Finally, the output result is predicted [32]. The Rectified Liner Uints (ReLU) activation function is used in the convolutional layer, and the Softmax activation function is used in the fully connected layer to classify data features. Eqs (1) and (2) express the activation functions.

$$ReLU = \max(0, x) \tag{1}$$

$$Softmax(x) = \frac{e^{x^i}}{\sum_i e^{x^i}} \tag{2}$$

In Eqs (1) and (2), the ReLU activation function outputs 0 or a positive number. When the

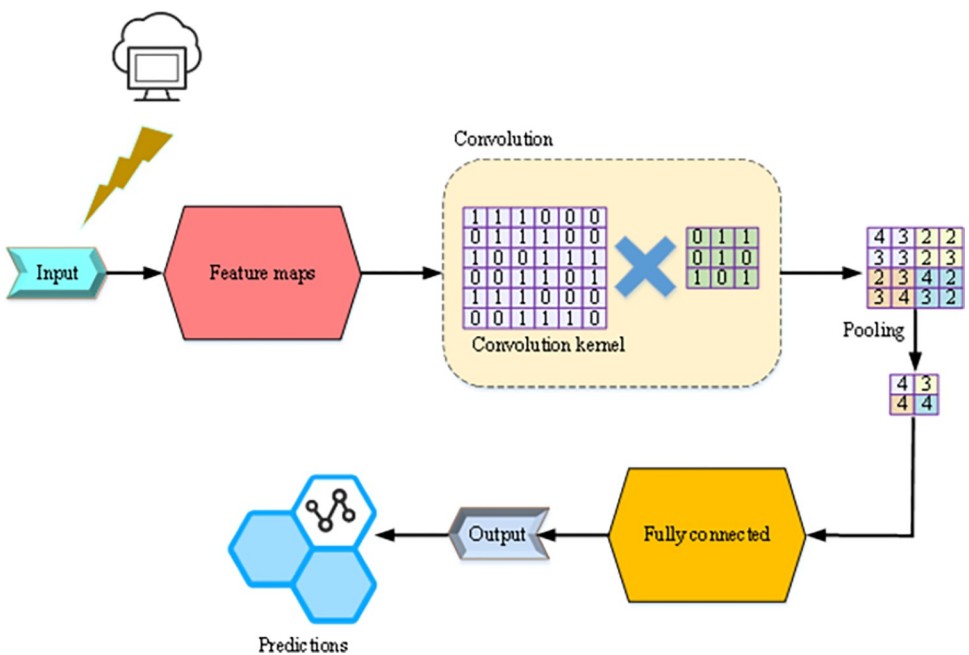

**Fig 1. CNN structure.**

Softmax(x) activation function handles more than two multi-classification problems, the class labels used need to have class membership. The activation functions used by different data features are the same for a particular convolutional layer, but the activation functions used by different convolutional layers can be different. In the process of convolution operation, let the input matrix be $x(i+m,j+n)$ and the convolution kernel be $w(m, n)$. Then, the above convolution process can be expressed as Eq (3).

$$Z(i,j) = (X * W)(i,j) = \sum_m \sum_n x(i + m, j + n)w(w, n) \tag{3}$$

In Eq (3), $Z(i,j)$ indicates the result of the convolution operation obtained by multiplying the convolution kernel $w(m, n)$ and the original data features. The convolution input is 6×6 a matrix, and the convolution kernel is a 3×3 data matrix. The convolution is performed by moving one data at a time. First, the upper left corner 1 of 6×6 input convolved with the convolution kernel. In other words, the element 4 obtained by multiplying and adding the elements of each position is the output matrix $Z(i,j)$. By analogy, the final output convolution matrix is obtained. Eq (4) describes the calculation process.

$$Z_{ij} = \sum_{k=1}^{n\_in}(X_k * W_k)(i,j) + b \tag{4}$$

In Eq (4), $n\_in$ represents the number of input matrices; $X_k$ refers to the $k$th input matrix; $W_k$ denotes the $k$th sub-convolution kernel in the convolution kernel; $Z_{ij}$ represents the value of the corresponding position element of the output matrix corresponding to the convolution operation. The data size after convolution is calculated according to Eq (5).

$$out = \frac{n + 2p - f}{s} + 1 \tag{5}$$

In Eq (5), $n$ stands for the data size extracted in the computer; $p$ signifies the number of padding data columns at the edge of the original data; $f$ represents the size of the convolution

kernel of the filter; *s* stands for the moving step size of the filter over the data features. This completes the convolution operation. Eq (6) describes the calculation output of the pooling operation in the CNN.

$$a^l = pool(a^{l-1}) \tag{6}$$

In Eq (6), *pool* represents the process of reducing the size of input data by pooling area *k* and pooling criteria; $a^{l-1}$ is an input tensor obtained by edge-filling the input data matrix by convolution. The fully connected layer calculates the output according to Eq (7).

$$a^l = \sigma(z^l) = \sigma(W^l a^{l-1} + b^l) \tag{7}$$

In Eq (7), *l* signifies the fully connected layer; $b^l$ refers to the threshold of the fully connected layer; σ stands for the activation function. Usually, Sigmoid and tanh are taken as the activation function. After several fully connected layers, the last layer is the output layer activated by Softmax. Eq (8) indicates the calculation of the output layer.

$$a^l = Softmax(z^l) = Softmax(W^l a^{l-1} + b^l) \tag{8}$$

Assume that $\alpha$ is the input gradient iteration parameter step size, and $\epsilon$ represents the maximum number of iterations and the stop iteration threshold. The output $\delta^{i,l}$ of the fully connected layer is obtained by calculating the loss function. Eqs (9) and (10) express the calculation in the fully connected layer after updating $W^l$ and $b^l$.

$$W^l = W^l - \alpha \sum_{i=1}^{m} \delta^{i,l} \left(a^{i,l-1}\right)^T \tag{9}$$

$$b^l = b^l - \alpha \sum_{i=1}^{m} \delta^{i,l} \tag{10}$$

## 3.2 Core idea of the PG algorithm

The PG algorithm mainly starts from the agent's policy for optimization. The strategy refers to the selection method of actions in a given state. The input of the neural network used in the PG algorithm is the corresponding current state of the agent. The output is the corresponding action (discrete space output probability of taking different actions and probability distribution of continuous space output). The PG algorithm models the policy function and then uses gradient descent to update the parameters of the network. However, there is no actual loss function in RL. The purpose of the PG algorithm is to maximize the expected value of the cumulative reward. Therefore, the expected value of the cumulative reward is used as the loss function, which is calculated through the Gradient Ascent algorithm [33]. According to this idea, the core idea of the PG algorithm can be expressed by Eq (11).

$$\pi = arg \ \max_{\pi} \ E_{\pi}\left[\sum_{t=0}^{\infty} r(s_t, a_t)\right] \tag{11}$$

In Eq (11), π represents the strategy, and *r* denotes the return value obtained at the moment *t*. The return value of the whole process is added to obtain the cumulative return value from the beginning to the end of the trajectory. The expectation $E_{\pi}$ of cumulative returns can be simply understood as taking the average of all possible processes. $s_t$ represents the state of the Neural Network obtained by the agent at the moment *t*. $a_t$ indicates the action of the agent at the moment *t*. A Parameterized Neural Network is used to express the strategy $\pi_{\theta}$. Then, the maximum expected return is calculated via Eq (12).

$$J(\pi_{\theta}) = \underset{\pi \sim \tau}{E} [\mathrm{R}(\tau)] \tag{12}$$

In Eq (12), $\tau$ represents the complete path from start to finish. The gradient ascent algorithm is used to find the maximum value, as presented in Eq (13).

$$\theta^* = \theta + \alpha \nabla J(\pi_\theta) \tag{13}$$

In Eq (13), $\theta^*$ signifies the maximization strategy objective, $\alpha$ stands for the optimal parameter for maximizing the payoff function, and $\theta$ represents the strategic objective. First, the gradient of the objective function is calculated, as shown in Eq (14).

$$\nabla J(\pi_\theta) = E_{\tau \sim \pi_\theta(\tau)}[\nabla_\theta log \pi_\theta(\tau) r(\tau)] \tag{14}$$

The log derivative technique is used in the above derivation. The derivative of $log\,x$ with respect to $x$ is $\frac{1}{x}$. Eq (14) can be derived from the derivation rule of the composite function. Furthermore, Eq (14) is decomposed. $\pi_\theta(\tau)$ is the strategy adopted by $\tau$ in the complete path from the beginning to the end. $\pi_\theta(\tau)$ is introduced to simplify $\nabla J(\pi_\theta)$ as Eq (15).

$$\nabla J(\pi_\theta) = E_{\tau \sim \pi_\theta(\tau)}\left[\sum_{t=1}^{T} \nabla_\theta log \pi_\theta(a_t|s_t)\left(\sum_{t=1}^{T} r(a_t, s_t)\right)\right] \tag{15}$$

So far, the gradient expression of the objective function has been obtained. However, the expected value cannot be calculated in the actual application process. This value can only be approximated by multiple sampling through the law of large numbers, as presented in Eq (16).

$$\nabla J(\pi_\theta) \approx \frac{1}{N}\sum_{i=1}^{N}\left[\sum_{t=1}^{T} \nabla_\theta log \pi_\theta(a_t|s_t)\left(\sum_{t=1}^{T} r(a_t, s_t)\right)\right] \tag{16}$$

In Eq (16), $N$ represents sampling $N$ times of different $\tau$, and $t = 1 \sim T$ represents the entire process of accumulating reward values from the beginning to the end. If $N = 1$, the gradient is updated every time a complete path is sampled. An improved Deep Deterministic Policy Gradient (DDPG) algorithm is used to realize the CDC task scheduling algorithm, aiming at the problems of high task scheduling cluster heterogeneity, significant cloud task volatility, and slow convergence speed of DRL algorithms [34]. Fig 2 reveals the implementation process of the DDPG algorithm.

The DDPG algorithm in Fig 2 is based on the Actor-Critic method. In terms of action output, the network fitting strategy function is used to directly output actions, which can cope with the output of continuous actions. The Critic in DDPG also outputs the $Q(s, a)$ value, but it is only the value corresponding to some sampling experience in the Reply buffer; no maximum search is required. $Q(s, a)$ is only used to train the Actor Policy Network. The final action is directly output by the network. There are also advantages of the Actor-Critic class method itself. The characteristics of the method can be combined based on the reward value and the strategy to directly give the best strategy, evaluate the candidate strategy through the critic, and constantly modify the Actor's strategy $(s, a)$ [35]. Eq 17 indicates the action expectation value of the agent in the Critic network.

$$Q(s_t, a_t|\theta^Q) = E[r(s_t, a_t) + \gamma Q(s_{t+1}, a_{t+1}|\theta^Q)] \tag{17}$$

In Eq (17), $s_t$ and $a_t$ are the state and action of the Critic network, respectively; $s_{t+1}$ and $a_{t+1}$ are the state and action of the Critic network at the next moment; $\theta^Q$ represents the parameter of the Critic network, mainly fitting values of $s_t$ and $a_t$; $\gamma$ refers to the network parameter.

Then, the action execution strategy of the Actor network in the $s_t$ state can be written as Eq (18).

$$Q(s_t, u(s_t)|\theta^Q) = E[r(s_t, u(s_t)) + \gamma Q(s_{t+1}, u(s_{t+1})|\theta^Q)] \tag{18}$$

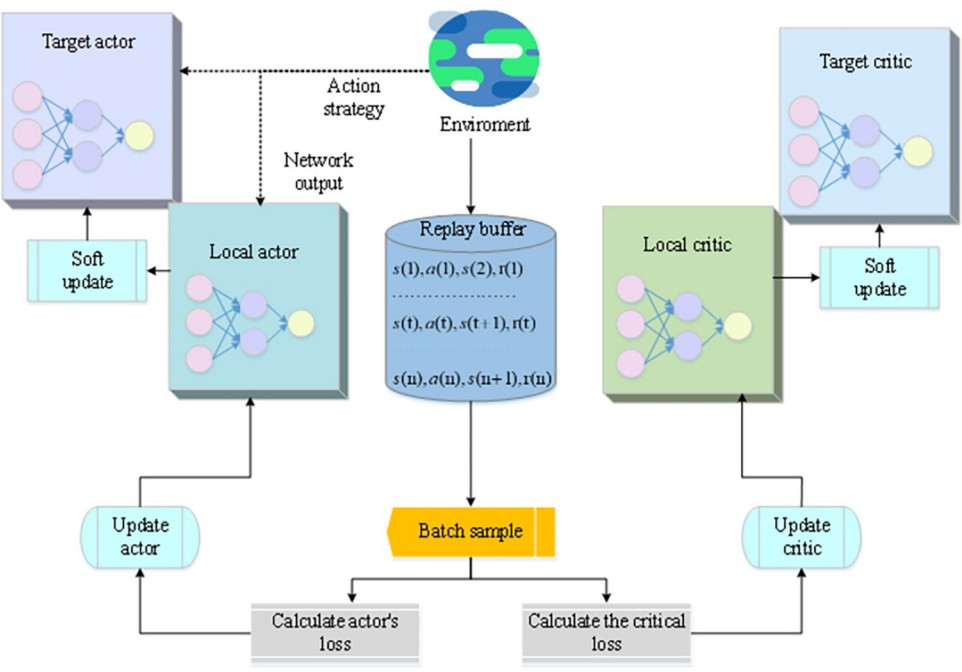

**Fig 2. Implementation process of the DDPG algorithm.**

As can be seen from Eq (17), the action execution policy of the Actor network is the action $u(s_t)$ executed by the action policy u of the agent in the Critic network.

## 3.3 Establishment of an CDC energy-saving model

Fig 3 displays the CDC data center.

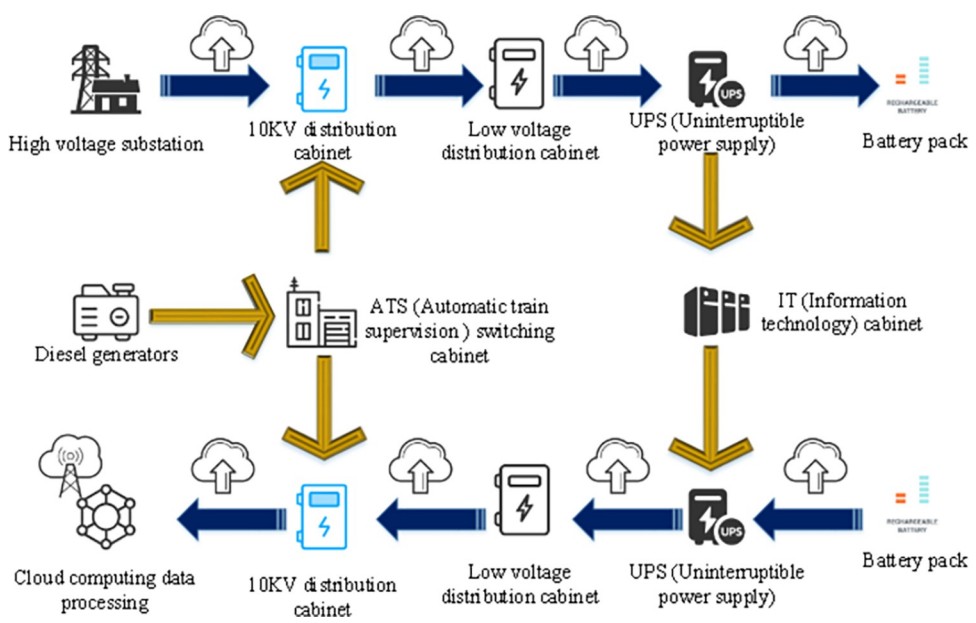

**Fig 3. CDC data center.**

Fig 3 shows that the CDC data center includes cabinets of different scales and the design of power equipment in the computer room. The terminal device provides an AC power supply plus a high voltage DC for the power supply, and the high voltage DC is directly embedded into the CDC device by hot swap [36]. In CDC data centers, the scale of CDC equipment continues to expand, and the overall energy consumption also increases. Therefore, the autonomous energy-saving management technology of CDC data centers should be optimized to control and reduce energy consumption. The traditional data center management methods and strategies are determined during deployment and operated following fixed patterns and processes. If these predefined management methods and strategies need to be adjusted, system administrators need to understand these methods and reconfigure the policy. System administrators need to understand and then reconfigure these predefined management methods and policies when needed. At the same time, data centers operating in cloud data centers usually reserve a considerable proportion of resource redundancy to meet the demand of peak load. Nevertheless, the actual load is mainly at a low level. In this case, many hardware devices do not provide effectual performance output while maintaining a high energy consumption.

The CNN uses the load and power consumption data in the system as the input. The network automatically generates an energy-saving strategy that conforms to the current system operating state and adjusts each component's operating mode. Under the premise of ensuring system stability and application performance requirements, the load of the entire monitored system is distributed more intensively to achieve a higher degree of energy consumption reduction. Fig 4 displays the CDC energy-saving workflow of CNNs.

Standardization and preprocessing of the original data (including data features and power consumption) are required before training a CNN. Then, the CNN model is trained to verify the trained model and check whether the basic error requirements are met. Finally, the whole model is applied to the cloud environment to independently complete the real-time data center

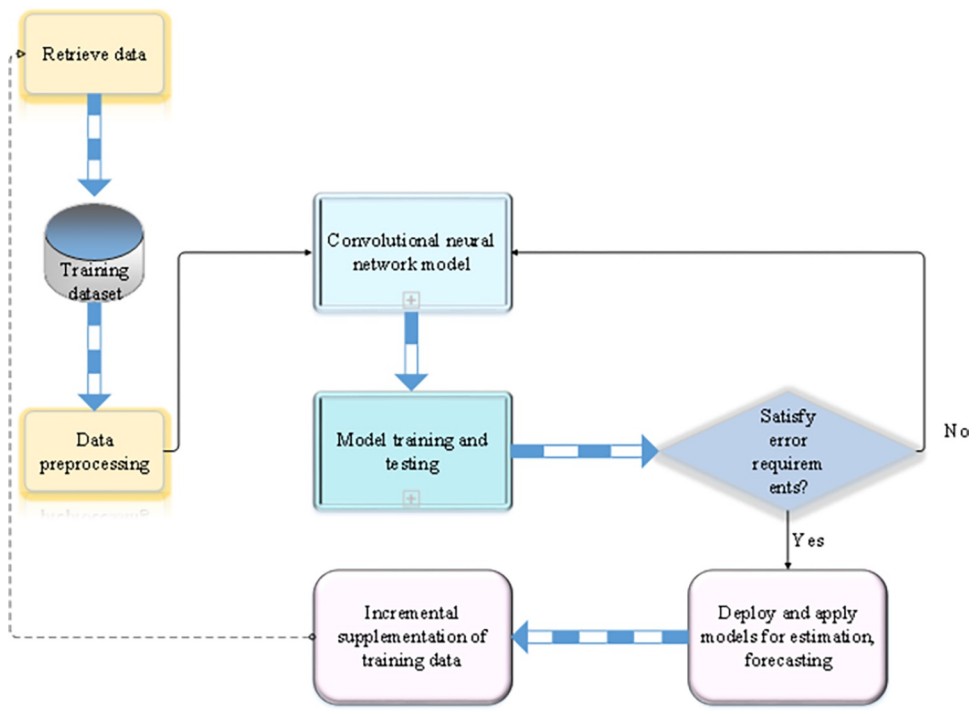

**Fig 4. CDC energy-efficient workflow of CNNs.**

data energy consumption. The incremental training in Fig 4 is suitable for situations where the hardware configuration changes or the type of load changes significantly. The training can be continued based on the current model by applying a new dataset to optimize the parameters.

## 3.4 Establishment of the CDC task scheduling model

When establishing a corresponding CDC energy-saving platform, it is necessary to allocate and manage the cloud data center's relevant resources according to users' needs to enhance the utilization rate of resources. The purpose of CDC task scheduling is to ensure that the task information submitted by users can be optimally scheduled to reach the maximum limit of the data processing capacity of the cloud data center. The main goals of CDC task scheduling include optimal span, service quality, economic principles, and load balancing. Fig 5 shows the task scheduling principle of the cloud data center.

The user submits the task to the cloud data center scheduling server (CDCSS). The CDCSS assigns the task to the appropriate virtual machine according to the task scheduling algorithm. A task queue is set in the CDCSS to store tasks submitted by users. The length of the task queue is unlimited and is used to store all different types of tasks submitted by different users. The three modules in the task scheduling algorithm, the status monitor, and the playback memory unit in the CDCSS share data. Fig 6 illustrates the cloud task scheduling strategy of the improved DDPG algorithm designed according to the PG algorithm.

## 3.5 Experiment preparation

Table 2 lists the experimental environment and configuration of this paper. According to the experimental environment configuration in Table 2, the Lawrence Livermore National

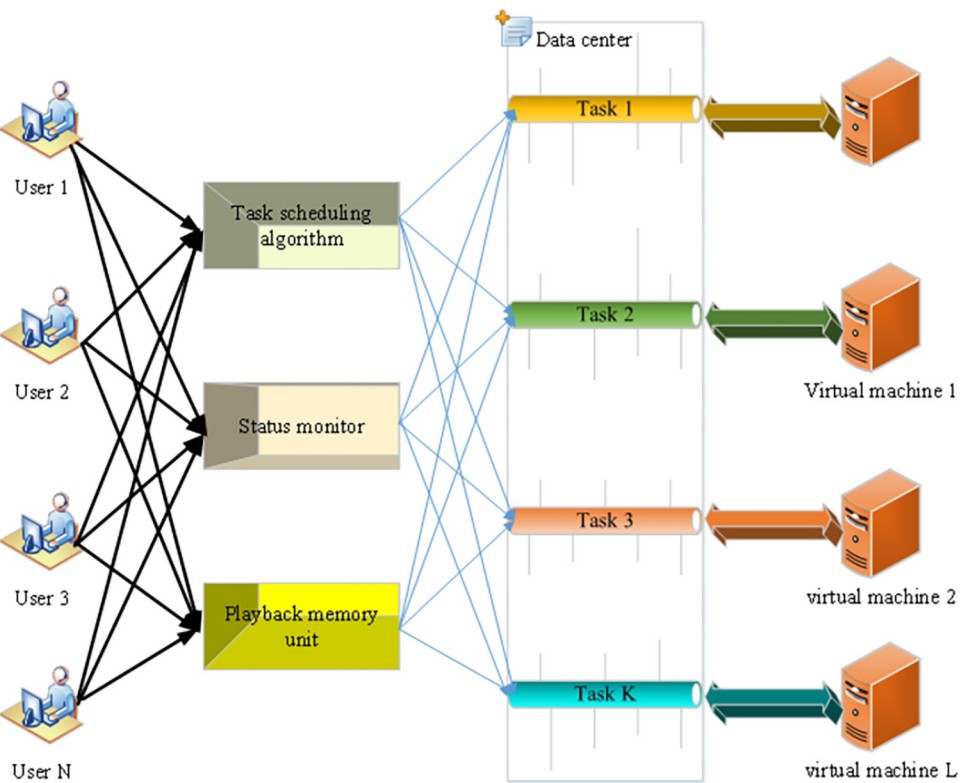

**Fig 5. Task scheduling principle of the cloud data center.**

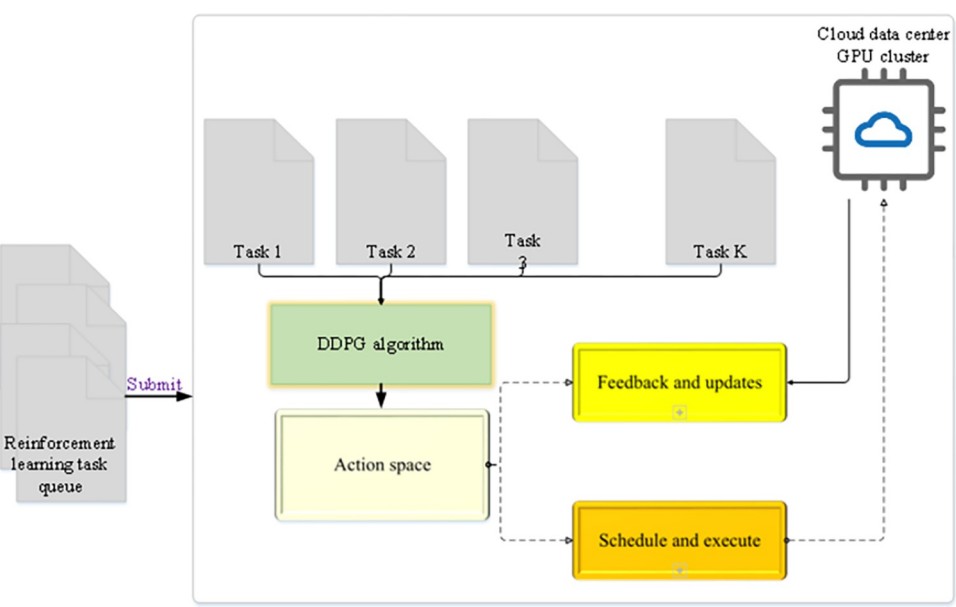

**Fig 6. Cloud task scheduling strategy based on the improved DDPG algorithm.**

Laboratory dataset is used to test the CDC energy-saving model and CDCSS. The Lawrence Livermore National Laboratory dataset includes detailed load data for large-scale CDC systems worldwide. The data link is Lawrence Livermore National Laboratory (llnl.gov). The data set link used by the article is: https://www.cs.huji.ac.il/labs/parallel/workload.

## 4. Results and discussions

### 4.1 Experimental results of the CDC energy-saving model

The CDC energy-saving model is tested and simulated in different scenarios. In Scenario 1, the requirements of the Plant Management System and the Virtual Mimicking System are set to 100, respectively; In Scenario 2, the requirements of the Plant Management System and the Virtual Mimicking System are set to 100 and 150, respectively; In Scenario 3, the requirements of the Plant Management System and the Virtual Mimicking System are set to 150 and 100, respectively. At the same time, the Elman Neural Network algorithm, ecoCloud algorithm, and CNN algorithm are applied to analyze the energy consumption and the number of virtual machine migrations in the cloud data center. Fig 7 presents the experimental results of the CDC energy-saving model with different algorithms.

As can be seen from Fig 7(A), CNN has the lowest energy consumption in the CDC energy-saving model compared with Elman and ecoCloud algorithms. In Scenario 1, the energy consumption of CNN is 30.13% lower than the Elman algorithm and 45.89% lower than the

**Table 2. Experimental environment and configuration.**

| | |
|---|---|
| Experimental hardware configuration | 2.12.1GHz Intel Xeon E5 CPU, 250GB Ramdom Access Memory |
| Operating system | Ubuntu 18.04 LTS |
| Software environment | Python 3.6 |
| Deep learning framework | PyTorch 1.3, Tensorflow 2.0 |
| Integrated development environment | JetBrains PyCharm 2019 |

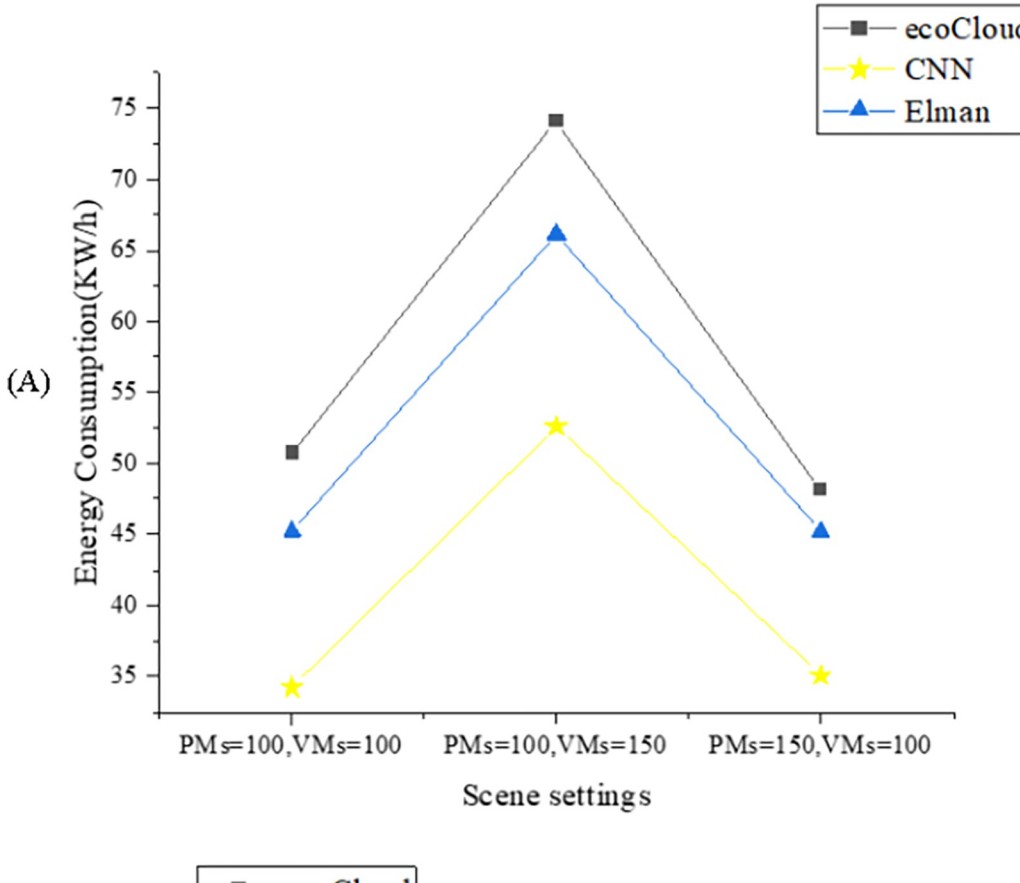

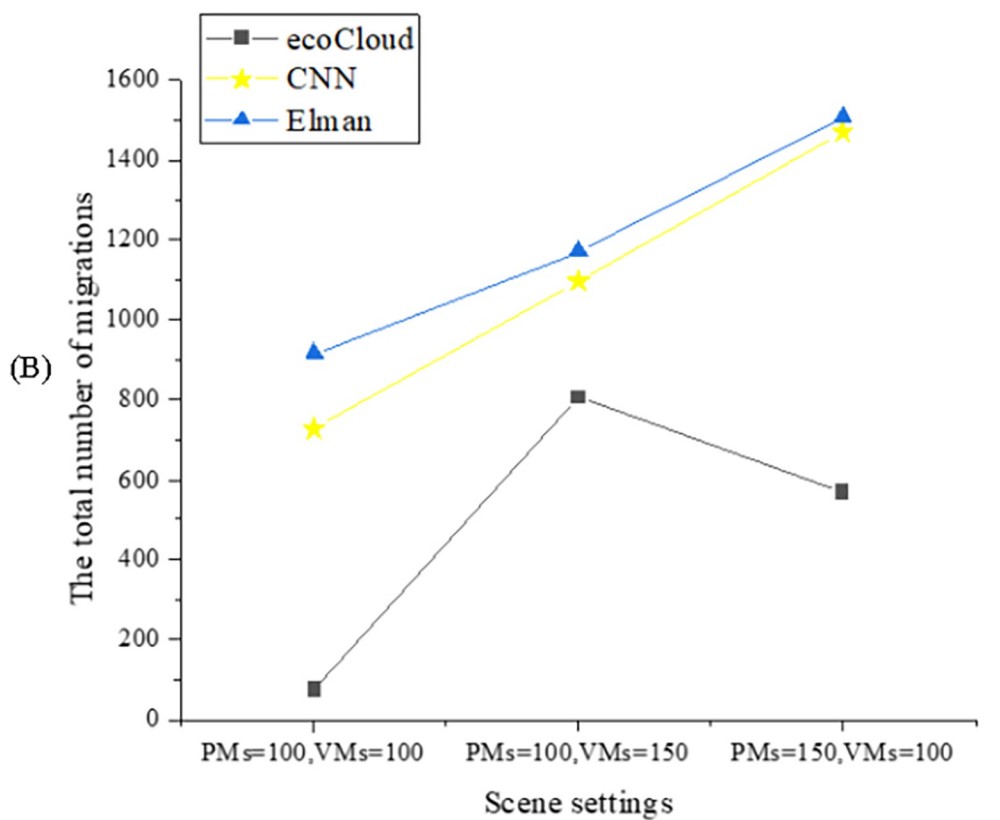

**Fig 7.** Experimental results of the CDC energy-saving models with different algorithms in different scenarios ((A) is the energy consumption; (B) is the number of virtual machine migrations).

ecoCloud algorithm. In Scenario 2, the energy consumption of CNN is 25.76% lower than the Elman algorithm and 40.80% lower than the ecoCloud algorithm. In Scenario 3, the energy consumption of CNN is 29.02% lower than the Elman algorithm and 37.48% lower than the ecoCloud algorithm. It is found that the energy consumption of the CNN algorithm in the CDC energy-saving model is better than that of the Elman algorithm and the ecoCloud algorithm. In Fig 7(B), the ecoCloud algorithm has the lowest number of virtual machine migrations in the three scenarios, with an average of 484. The CNN and Elman algorithms have similar virtual machine migration times in the three scenarios; the average migrations are 1,097 times and 1,199 times. However, according to the CDC principle of ecoCloud, most of the server energy consumption of the ecoCloud algorithm is the resource consumption when the server is idle. Therefore, considering the execution of the CNN algorithm and Elman algorithm, it can be found that compared with the Elman algorithm, the CNN algorithm reduces the number of VM migrations in CDC energy-saving model by 9.30%.

## 4.2 Experiment results of CDC task scheduling strategy

The command response time of the Random, Earliest algorithm, Round-Robin (RR), DQN, and DDPG algorithms used in the cloud data center are compared in different scenarios. Fig 8 provides the task scheduling effect of different algorithms.

Combined with the response time of CDC task scheduling in three scenarios in Fig 8, it is found that the DDPG algorithm has the highest instruction response time in different scenarios. The average response time of the DDPG algorithm in the three scenarios is 141. With the difficulty of setting the model scene and the change of the submission times, the average response time gap between the DDPG algorithm and RR algorithm gradually becomes smaller. The average response time of the Random algorithm is 111, that of the Earliest algorithm is 129, that of the RR algorithm is 137, and that of the DQN algorithm is 126. Therefore, the DDPG algorithm performs the best in the task scheduling strategy of the cloud data center, and the DRL algorithm DQN algorithm performs poorly.

## 5. Discussion

The work shows that the CNN algorithm outperforms the Elman algorithm and ecoCloud algorithm for energy consumption in cloud data energy-saving models. Qu et al. (2022) established an energy consumption model based on the Elman algorithm in studying energy consumption in cloud computing environments and energy-saving scheduling tasks. They found that the accuracy of the energy consumption model based on the Elman algorithm is higher than that of the multiple linear regression model [37]. They proved the correctness of this paper to study the comparison method between the CNN algorithm and the Elman algorithm in CDC energy consumption. The command response time of the DDPG algorithm is the highest under different scenarios. The average response time of the DDPG algorithm under three scenarios is 141, with a change in the difficulty of setting up the model scenarios. The average response time gap between the DDPG algorithm and the RR algorithm gradually becomes smaller with the change in the number of test submissions for algorithms with high-setting scenarios. Shi et al. (2020) proposed efficient cloud task scheduling algorithms, the DQN algorithm and the DDPN algorithm, respectively, based on DRL. The results showed that the DDPN algorithm converged faster and outperformed the traditional algorithms in

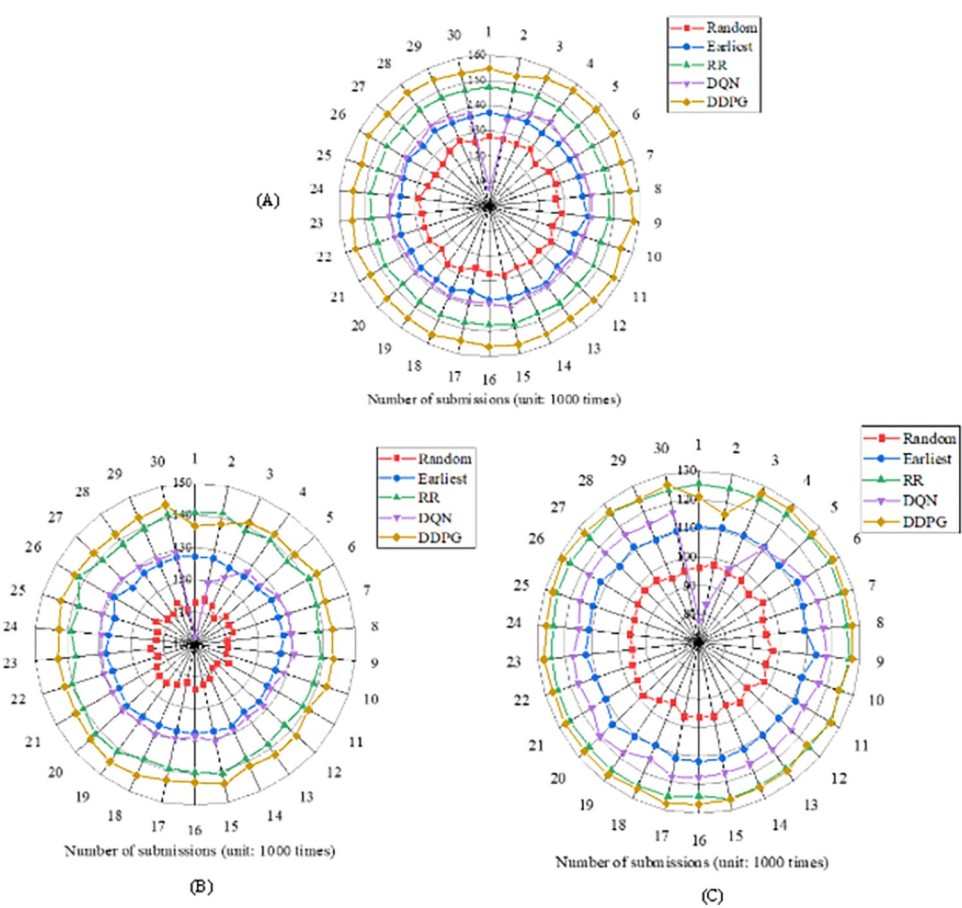

**Fig 8.** Task scheduling response time of different algorithms in the cloud data center in different scenarios ((A) is Scenario 1; (B) is Scenario 2; (C) is Scenario 3).

terms of optimization metrics such as command response time ratio and load balancing [38]. This study proves the feasibility of the results of this work.

## 6. Conclusion

This work utilizes a CNN to adjust each hardware resource module's power and operation mode in the cloud data center. In addition, the energy consumption requirements of a single hardware resource module are reduced as much as possible to ensure system stability and application performance. Under the premise of performance and application performance requirements, the CDC monitoring system's load is distributed more centrally so that some CDC devices can enter standby, power off, or other equivalent states to reduce energy consumption. At the same time, the improved PG algorithm is used to analyze the time response of the task scheduling strategy of the cloud data center. It reflects the energy-saving effect of the task scheduling time of the cloud data center from the side, dramatically saving the data computing response time. Relevant experiments are carried out using the Lawrence Livermore National Laboratory data set. The experiments suggest that the energy consumption of the CNN algorithm in the CDC energy-saving model is better than that of the Elman algorithm and the ecoCloud algorithm. The CNN and Elman algorithms have similar virtual machine migration times in the three scenarios, with an average of 1,097 and 1,199 times, respectively.

Compared with the Elman algorithm, the CNN algorithm reduces the number of virtual machine migrations in the CDC energy-saving model by 9.30%. The command response time of the DDPG algorithm in different scenarios is the highest. The average response time of the DDPG algorithm in the three scenarios is 141. Furthermore, it is found that the response time of the algorithm increases with the number of test submissions with the difficulty of the model scenario setup. The difference between the average response time of the DDPG algorithm and the RR algorithm gradually becomes smaller.

This paper studies the neural network algorithms in the scenario of high heterogeneity of CDC clusters and significant task volatility. The algorithm proposed here has good convergence, greatly improves the command response time ratio of tasks, and promotes cluster load balancing. It realizes the computing energy-saving strategy of the cloud data center. However, there are still some shortcomings. This work highlighted the practicability of the CDC energy-saving model mainly from the energy consumption of computing equipment in the cloud data center but did not consider other energy-saving factors. Future research will comprehensively consider the energy-saving factors of cloud data center computing for a comprehensive CDC energy-saving strategy.

## Supporting information

**S1 Data.**
(ZIP)

**S2 Data.**
(XLSX)

## Author Contributions

**Conceptualization:** Dexian Yang.

**Data curation:** Dexian Yang.

**Formal analysis:** Dexian Yang.

**Funding acquisition:** Jiong Yu.

**Investigation:** Jiong Yu.

**Methodology:** Jiong Yu.

**Project administration:** Xusheng Du.

**Resources:** Xusheng Du.

**Software:** Xusheng Du.

**Supervision:** Zhenzhen He.

**Validation:** Zhenzhen He.

**Visualization:** Zhenzhen He.

**Writing – original draft:** Ping Li.

**Writing – review & editing:** Ping Li.

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
