## [Decision Letter · Decision Letter 0]

6 Oct 2022

PONE-D-22-22823Energy Saving Strategy of Cloud Data Computing Based on Convolutional Neural Network and Policy Gradient AlgorithmPLOS ONE

Dear Dr. Yang,

Thank you for submitting your manuscript to PLOS ONE. After careful consideration, we feel that it has merit but does not fully meet PLOS ONE’s publication criteria as it currently stands. Therefore, we invite you to submit a revised version of the manuscript that addresses the points raised during the review process.

We look forward to receiving your revised manuscript.

Kind regards,

Osamah Shihab Albahrey

Academic Editor

PLOS ONE

Journal Requirements:

"This research was supported by the National Natural Science Foundation of China under Grant Nos. 61862060."

"This research was supported by the National Natural Science Foundation of China under Grant Nos. 61862060."

Additional Editor Comments:

The reviewers have commented on the paper. This paper could be published if you revised it carefully considering all given comments.

Reviewers' comments:

Reviewer's Responses to Questions

**Comments to the Author**

1. Is the manuscript technically sound, and do the data support the conclusions?

Reviewer #1: Yes

Reviewer #2: Yes

2. Has the statistical analysis been performed appropriately and rigorously? 

Reviewer #1: Yes

Reviewer #2: N/A

3. Have the authors made all data underlying the findings in their manuscript fully available?

Reviewer #1: No

Reviewer #2: Yes

4. Is the manuscript presented in an intelligible fashion and written in standard English?

Reviewer #1: Yes

Reviewer #2: Yes

5. Review Comments to the Author

Reviewer #1: This work combined Deep Reinforcement Learning and neural networks to reduce the energy consumption of task scheduling in Cloud Data Computing data centers. The work sounds good and can be published after minor revision.

Comments:

-Lack of references in the introduction section. Many issues mentioned without reference.

-For reader convenience, I suggest that the notations should be listed with one table with their meanings

-The data in the excel sheet need to be specified more clearly.

-Please add a link (URL) to the dataset which you used.

Reviewer #2: My Comments for the work are as follows,

==== INTRODUCTION ====

The authors need to better explain the context of this research, including why the research problem is important.

The introduction should clearly explain the key limitations of prior work that are relevant to this paper.

Contributions should be highlighted more. It should be made clear what is novel and how it addresses the limitations of prior work.

==== RELATED WORK ====

The authors should explain clearly what the differences are between the prior work and the solution presented in this paper.

The authors should add a table that compares the key characteristics of prior work to highlight their differences and limitations. The authors may also consider adding a line in the table to describe the proposed solution.

==== METHOD ====

The authors should first give an overview of their solution before explaining the details.

It is important to clearly explain what is new and what is not in the proposed solution. If some parts are identical, they should be appropriately cited and differences should be highlighted.

The authors should add proof(s) of the properties, theorem or lemmas contained in the paper.

==== EXPERIMENTS ====

There is not enough discussion of the experimental results.

6. PLOS authors have the option to publish the peer review history of their article (what does this mean?). If published, this will include your full peer review and any attached files.

Reviewer #1: No

Reviewer #2: No

---

## [Author Response · Author response to Decision Letter 0]

16 Oct 2022

Reviewer #1: This work combined Deep Reinforcement Learning and neural networks to reduce the energy consumption of task scheduling in Cloud Data Computing data centers. The work sounds good and can be published after minor revision.

Comments:

-Lack of references in the introduction section. Many issues mentioned without reference.

Reply: Thank you for your comment. According to the introduction, supplementary references have been provided. See References 1, 3, 4, 7, 8, 9, 11 for details.

-For reader convenience, I suggest that the notations should be listed with one table with their meanings

Reply: Thank you for your comment. Table 1 has been cited to illustrate the characteristics of relevant scholars' research.

-The data in the excel sheet need to be specified more clearly.

Reply: Thank you for your comment. For a detailed description of the data in excel, see the raw file.

-Please add a link (URL) to the dataset which you used.

Reply: Thank you for your comment. This article uses links to datasets explained in Section 3.5.

Reviewer #2: My Comments for the work are as follows,

==== INTRODUCTION ====

The authors need to better explain the context of this research, including why the research problem is important.

Reply: Thank you for your comment. The research background has been explained in detail in the second paragraph of the introduction section. In addition, for the research purpose of this article, it has been explained in the third paragraph of the introduction section. 

The introduction should clearly explain the key limitations of prior work that are relevant to this paper.

Reply: Thank you for your comment. In the introduction section, the differences between the research of this paper and previous research work, as well as the shortcomings of previous research on energy planning and energy management have been explained, leading to the research of this paper. See the fourth paragraph of the introduction section.

Contributions should be highlighted more. It should be made clear what is novel and how it addresses the limitations of prior work.

Reply: Thank you for your comment. The research contribution of this paper has been clarified in the introduction section, and the innovation of this paper has been explained, as well as the advantages of cloud data computing task scheduling model based on policy gradient algorithm in this paper, which solves the energy consumption problem of cloud data computing. Please refer to the fourth paragraph of the introduction section for details.

==== RELATED WORK ====

The authors should explain clearly what the differences are between the prior work and the solution presented in this paper.

Reply: Thank you for your comment. The differences between this study and previous studies have been explained in detail. See the second paragraph of Section 2.2 for details.

The authors should add a table that compares the key characteristics of prior work to highlight their differences and limitations. The authors may also consider adding a line in the table to describe the proposed solution.

Reply: Thank you for your comment. A table has been added in Section 2.2 to explain the characteristics of previous research methods and the advantages of this research method. See Table 1 for details.

==== METHOD ====

The authors should first give an overview of their solution before explaining the details.

Reply: Thank you for your comment. The research process of this paper has been explained in the research method section. Please refer to the first paragraph of Section 3.1 for details.

It is important to clearly explain what is new and what is not in the proposed solution. If some parts are identical, they should be appropriately cited and differences should be highlighted.

Reply: Thank you for your comment. References have been added to the first paragraph of Section 3.1 to highlight the innovation of this research method. Please refer to the first paragraph of Section 3.1 for details and references 29, 30.

The authors should add proof(s) of the properties, theorem or lemmas contained in the paper.

Reply: Thank you for your comment. In this paper, the process of convolutional neural network algorithm has been explained in Equations 1 to 10, and the derivation process of strategy gradient algorithm has been supplemented in Section 3.2. Please refer to current Equations 12 to 16 for details.

==== EXPERIMENTS ====

There is not enough discussion of the experimental results.

Reply: Thank you for your comment. The research results of this paper have been discussed based on the research of scholars. Please refer to "5. Discussion" for details.

---

## [Decision Letter · Decision Letter 1]

16 Nov 2022

PONE-D-22-22823R1Energy Saving Strategy of Cloud Data Computing Based on Convolutional Neural Network and Policy Gradient AlgorithmPLOS ONE

Dear Dr. Yang,

Thank you for submitting your manuscript to PLOS ONE. After careful consideration, we feel that it has merit but does not fully meet PLOS ONE’s publication criteria as it currently stands. Therefore, we invite you to submit a revised version of the manuscript that addresses the points raised during the review process.

We look forward to receiving your revised manuscript.

Kind regards,

Osamah Shihab Albahrey

Academic Editor

PLOS ONE

Journal Requirements:

Additional Editor Comments:

The reviewers have commented on the paper. They feel your paper should undergo a second round of minor corrections. Please revise them accordingly.

Reviewers' comments:

Reviewer's Responses to Questions

**Comments to the Author**

1. If the authors have adequately addressed your comments raised in a previous round of review and you feel that this manuscript is now acceptable for publication, you may indicate that here to bypass the “Comments to the Author” section, enter your conflict of interest statement in the “Confidential to Editor” section, and submit your "Accept" recommendation.

Reviewer #1: (No Response)

2. Is the manuscript technically sound, and do the data support the conclusions?

Reviewer #1: Yes

3. Has the statistical analysis been performed appropriately and rigorously? 

Reviewer #1: (No Response)

4. Have the authors made all data underlying the findings in their manuscript fully available?

Reviewer #1: No

5. Is the manuscript presented in an intelligible fashion and written in standard English?

Reviewer #1: (No Response)

6. Review Comments to the Author

Reviewer #1: Second revision

Lack of references in the introduction section. Many issues mentioned without reference. (Solved )

-For reader convenience, I suggest that the notations should be listed with one table with their meanings

Author answer: Thank you for your comment. Table 1 has been cited to illustrate the characteristics of relevant scholars' research.

-By list of notations I mean something like this:

http://www.nlpr.ia.ac.cn/users/szli/MRF_Book/Head-Tail/node162.html

-Please add a link (URL) to the dataset which you used.

Author answer: Thank you for your comment. This article uses links to datasets explained in Section 3.5.

There is no links to data in section 3.5.

7. PLOS authors have the option to publish the peer review history of their article (what does this mean?). If published, this will include your full peer review and any attached files.

Reviewer #1: No

---

## [Author Response · Author response to Decision Letter 1]

21 Nov 2022

There is no links to data in section 3.5.

Reply: Thank you for your comments. The article dataset link has been added. See Section 3.5 of the article for details.

---

## [Editor Report · Decision Letter 2]

12 Dec 2022

Energy Saving Strategy of Cloud Data Computing Based on Convolutional Neural Network and Policy Gradient Algorithm

PONE-D-22-22823R2

Dear Dr. Yang,

We’re pleased to inform you that your manuscript has been judged scientifically suitable for publication and will be formally accepted for publication once it meets all outstanding technical requirements.

Kind regards,

Osamah Shihab Albahrey

Academic Editor

PLOS ONE

Additional Editor Comments (optional):

Dear authors,

You have successfully revised the given comments properly.
---

## [Editor Report · Acceptance letter]

19 Dec 2022

PONE-D-22-22823R2 

Energy Saving Strategy of Cloud Data Computing Based on Convolutional Neural Network and Policy Gradient Algorithm 

Dear Dr. Yang:

I'm pleased to inform you that your manuscript has been deemed suitable for publication in PLOS ONE. Congratulations! Your manuscript is now with our production department. 

Kind regards, 

on behalf of

Dr. Osamah Shihab Albahrey 

Academic Editor

PLOS ONE